

# Detection of a *Yersinia pestis* gene homologue in rodent samples

Timothy A. Giles[1], Alex D. Greenwood[2,3], Kyriakos Tsangaras[2,4], Tom C. Giles[5], Paul A. Barrow[1], Duncan Hannant[1], Abu-Bakr Abu-Median[1] and Lisa Yon[1]

[1] School of Veterinary Medicine and Science, University of Nottingham, Loughborough, Leicestershire, United Kingdom
[2] Department of Wildlife Diseases, Leibniz Institute for Zoo and Wildlife Research, Berlin, Germany
[3] Department of Veterinary Medicine, Freie Universität Berlin, Berlin, Germany
[4] Department of Translational Genetics, Cyprus Institute of Neurology and Genetics, Nicosia, Cyprus
[5] The Advanced Data Analysis Centre, University of Nottingham, Leicestershire, United Kingdom

## ABSTRACT

A homologue to a widely used genetic marker, *pla,* for *Yersinia pestis* has been identified in tissue samples of two species of rat (*Rattus rattus* and *Rattus norvegicus*) and of mice (*Mus musculus* and *Apodemus sylvaticus*) using a microarray based platform to screen for zoonotic pathogens of interest. Samples were from urban locations in the UK (Liverpool) and Canada (Vancouver). The results indicate the presence of an unknown bacterium that shares a homologue for the *pla* gene of *Yersinia pestis,* so caution should be taken when using this gene as a diagnostic marker.

# INTRODUCTION

*Yersinia pestis* is the causative agent of plague in humans and, in the absence of antimicrobial therapy, the mortality rate can approach 100%. In large parts of the world the threat from *Y. pestis* has declined substantially over time as a result of improvements in living conditions and in public health, including improved rodent control and antibiotics. However, a plague outbreak following the release of a biological weapon is a potential risk. The presence of *Y. pestis* in small rodent populations in which it is endemic (*Ziwa et al., 2013*; *Eppinger et al., 2009*; *Biggins & Kosoy, 2001*) can cause human fatalities as a result of zoonotic transmission (*International Society for Infectious Diseases, 2014*).

The black rat (*Rattus rattus*) has been a major host of *Y. pestis* for centuries and can be a reservoir for numerous other pathogens. Although most mammalian species can be infected experimentally with *Y. pestis*, many species fail to develop the high bacteraemia that is necessary to infect the flea vectors. The majority of mammalian species are therefore likely to be dead end hosts (*Eisen & Gage, 2009*).

Molecular methods, and in particular PCR, have been widely used to identify *Y. pestis* in tissue samples and the plasminogen activator/coagulase (*pla*) gene, located on the pPCP1 plasmid has been used as a target in many studies (*Loïez et al., 2003*; *Raoult et al., 2000*; *Zhang et al., 2013*; *Ziwa et al., 2013*). The *pla* gene is commonly used because it has a high copy number in *Y. pestis* (186 per bacterium) and can be detected relatively easily (*Parkhill*

Corresponding author
Timothy A. Giles,
timothy.giles@nottingham.ac.uk

**Table 1  *Y. pestis* primers used to prepare baits for Illumina Miseq sequencing.**

| Primer Name | Primer sequence | Target | Reference |
|---|---|---|---|
| F1 | CAGTTCCGTTATCGCCATTGC | *caf1* | *Norkina et al. (1994)* |
| F2 | TATTGGTTAGATACGGTTACGGT | | |
| Ypfur1 | GAAGTGTTGCAAAATCCTGCG | *fur* | *Hinnebusch, Fischer & Schwan (1998)* |
| Ypfur2 | AGTGACCGTATAAATACAGGC | | |
| YPtoxU | AGGACCTAATATGGAGCATGAC | *Ymt* | *Riehm et al. (2011)* |
| YPtoxUR | CGTGATTACCAGGTGCAACA | | |

*et al., 2001*). The PcP plasmid, and the *pla* gene in particular, is involved in transmission of *Y. pestis* (*Broekhuijsen et al., 2003*). The protein encoded by the *pla* gene induces fibrinolysis and degrades the extracellular matrix and basement membranes, these activities are thought to disrupt the host's ability to contain the bacteria (*Sebbane et al., 2006*). The acquisition by *Y. pestis* of PcP, and of another virulence plasmid, the pMT, is thought to have contributed to the evolutionary transformation of *Y. pestis* from the mainly gut-associated *Yersinia enterocolitica*, into a highly host-adapted mammalian blood-borne pathogen (*Eppinger et al., 2010*).

## RESULTS AND DISCUSSION

Probes specific to *Y. pestis* hybridised with samples from a subset of each of the rodent species tested (12/33 *R. rattus*, 48/834 *R. norvegicus*, 3/163 *A. sylvaticus*, 2/35 *M. musculus*) giving a total of 65/1065 samples (6.1%) which tested positive on the array. However, none of the generic *Yersinia* probes hybridised in those samples for which a positive signal was recorded for the *Y. pestis* specific probes.

Further testing was then carried out at the University of Nottingham, including real-time PCR which targeted another region of the *Y. pestis* genome, the *caf1* gene, for which primers used were identified from the literature (*Janse et al., 2010*). A subset (23 samples) of the array-positive samples was tested further with primers for *pla* and *caf1*. Of these samples, 12 were positive for the *pla* gene and all were negative for the *caf1* gene. A total of 30 array-positive samples were also sent to colleagues in Berlin for further analysis by in solution-based sequence hybridisation, as described previously (*Tsangaras et al., 2014*). Briefly, a DNA extract from the samples was fragmented and an aliquot was used to produce illumina libraries following a custom protocol (*Meyer & Kircher, 2010*). PCR amplicons from *Y. pestis* genes were used to enrich specific target DNA sequences in the rodent samples, the genes and primers used to make the baits are shown in Table 1. The enriched samples were then sequenced using an illumina Miseq. The results indicated that the majority of reads aligned with the rat genome, and only a small number of reads aligned with the *pla* gene. No other reads mapped to any other gene from *Y. pestis*.

Although a homologue to the *pla* gene has previously been reported in bacteria found in *R. rattus* and *R. norvegicus* from the Netherlands (*Janse, Hamidjaja & Reusken, 2013*), this is the first time, to the author's knowledge, that the bacterial homologue has been reported in *M. musculus* and *A. sylvaticus*. The potential discovery of a *pla* gene homologue in other

rodent species, and on another continent than the species and locations in which it has previously been reported, suggests that the homologue could be more widely distributed than previously thought and may cause difficulties in accurate *Y. pestis* detection. The results found here supports other work which suggests that markers other than the *pla* gene should be included to help avoid false positive results when screening for *Y. pestis*, as has been stressed by *Janse, Hamidjaja & Reusken (2013)*. This was recently confirmed by Hänsch et al., as they found evidence that the pla gene is present in some strains of *Escherichia coli* and *Citrobacter koseri* (*Hänsch et al., 2015*). It is not clear why the homologue was present in a larger percentage of *R. rattus* samples than in the other species tested; perhaps *R. rattus* carries more *E. coli* or *C. koseri*, but this is something that needs to be investigated further.

## MATERIALS AND METHODS

This work was part of a EU project (FP7 WildTech) to develop and use a microarray to detect zoonotic pathogens in rodent tissues. A sequence of the pPCP1 plasmid of the *Y. pestis* genome (CP000310.1) was obtained from the NCBI database for microarray probe design. Probes were designed using two publicly available software packages: OligoWiz (http://www.cbs.dtu.dk/services/OligoWiz/) and Unique Probe Selector (http://array.iis.sinica.edu.tw/ups/). All probes were checked for suitability using an *in silico* BLAST analysis. The results of the *in silico* analysis at the time indicated that the probe sequences were specific to *Y. pestis* and no cross-hybridisation should occur with eukaryotic or prokaryotic species. Primers were designed using the software Primer3 (http://bioinfo.ut.ee/primer3-0.4.0/). The sequence of each oligonucleotide probe specific to *Y. pestis* is given in Table 2. During the confirmatory testing, both real-time PCR and end-point PCR were used. The primers used in standard end-point PCR and real-time PCR are shown in Table 3. These probes were evaluated thoroughly for specificity using reference samples of genomic DNA from *Y. pestis* NCTC5923 Type strain and non-related pathogens before screening took place. The microarray platform used was the ArrayStrip from Alere Technologies GmbH (Jena, Germany).

Four different rodent species (*R. rattus, R. norvegicus, Mus musculus* and *Apodemus sylvaticus*) were screened for a number of zoonotic pathogens. Tissue samples were obtained from Vancouver (Canada), Liverpool (UK), and Lyon (France) as part of other studies. Automated nucleic acid extraction was performed on the samples using the QIAcube (Qiagen, Hilden, Germany) and the kit (Cador Pathogen Mini Kit; Qiagen, Hilden, Germany). Liver, kidney and lung tissues were available from each rodent sampled from Vancouver and Lyon, and extracted nucleic acid from each tissue was pooled to make a single sample per individual animal which was tested on the array. Only liver and kidney samples were available from the rodents sampled from Liverpool, and again, extracted nucleic acid was pooled to make a single sample. Figure 1 depicts the sequence enrichment and microarray hybridisation used in this study.

**Table 2  *Y. pestis* specific oligonucleotide probes used on the Alere ArrayStrip.**

| Probe | Sequence (5′–3′) | Pathogen | Gene target | Position[a] |
|---|---|---|---|---|
| Y.pestis_Owiz_117 | TACAGATCATATCTCTCTTTTCATCCTCCCCTAGCGGGGAGGATGTCTGTGGAAAGGAGG | *Y. pestis* | pPCP1 | 8781–8840 |
| Y.pestis_Owiz_120 | TGTTGTCCGCTAGGACGATGCGATTTCGGTTATTATTCAGAATGTCTTCGTTCTCTTTC | *Y. pestis* | pPCP1 | 6626–6684 |
| Y.pestis_Owiz_121 | TGTCCGGGAGTGCTAATGCAGCATCATCTCAGTTAATACCAAATATATCCCCTGACAGC | *Y. pestis* | pPCP1 | 7878–7936 |
| Y.pestis_Owiz_127 | GTGGAGATTCTGTCTCTATTGGCGGAGATGCTGCCGGTATTTCCAATAAAAATTATACTG | *Y. pestis* | pPCP1 | 8688–8747 |
| Y.pestis_Owiz_129 | GAATCGCGCCCGGATATGTTTTAACGCGATTTTCAGACTCAGACAAATTCAGCAGAAT | *Y. pestis* | pPCP1 | 9990–10047 |
| Y.pestis_Owiz_147 | TCGCTGGCTAAAAAGTACCATCCACATGCTCAACCCTATAACCTGTAGCTTACCCCAC | *Y. pestis* | pPCP1 | 9583–9640 |
| YpestisUPS_785 | AATAGGTTATAACCAGCGCTTTTCTATGCCATATATTGGACTTGCAGGCCAGTATCGCAT | *Y. pestis* | pPCP1 | 8392–8451 |
| YpestisUPS_786 | AATGATGAGCACTATATGAGAGATCTTACTTTCCGTGAGAAGACATCCGGCTCACGTTAT | *Y. pestis* | pPCP1 | 8510–8569 |
| YpestisUPS_787 | TAAATTCAGCGACTGGGTTCGGGCACATGATAATGATGAGCACTATATGAGAGATCTTAC | *Y. pestis* | pPCP1 | 8479–8538 |
| Y.pestisUPS_788 | AGCCCGACCACTGCGCCTTATCCGGTAACTATCGTCTTGAGTCCAACCCGGTAAGACACG | *Y. pestis* | pPCP1 | 4977–5036 |
| YpestisUPS_789 | TCATCCTCCCCTAGCGGGGAGGATGTCTGTGGAAAGGAGGTTGGTGTTTGACCAACCTTC | *Y. pestis* | pPCP | 8801–8860 |
| YpestisUPS_790 | AAAGGACAGCATTTGGTATCTGTGCTCCACTTAAGCCAGCTACCACAGGTTAGAAAGCCT | *Y. pestis* | pPCP | 5129–5188 |
| YpestisUPS_791 | AAGGAGTGCGGGTAATAGGTTATAACCAGCGCTTTTCTATGCCATATATTGGACTTGCAG | *Y. pestis* | pPCP | 8379–8438 |
| YpestisUPS_792 | TTTGTACCGAGAACCTTTCACGGTATCGGCATATGGCCTGGGTAACTCAGGTCCGTAAAC | *Y. pestis* | pPCP | 9451–9510 |

**Notes.**
[a]The nucleotide position of each probe is based on the CP000310.1 *Yersinia pestis* Antiqua plasmid pPCP.

Giles et al. (2016), *PeerJ*, DOI 10.7717/peerj.2216

**Table 3** *Y. pestis* **specific primers for standard end-point PCR and real-time PCR.**

| Forward Primer | Sequence (5′–3′) | Reverse primer | Sequence (5′–3′) | Probe | Sequence (5′–3′) | Gene | Position |
|---|---|---|---|---|---|---|---|
| Y.pes/pPCP/8374/F | CCCGAAAGGAG TGCGGGTAA | Y.pes/pPCP/8902/R | CGCCCCGTCATT ATGGTGAA | N/A | N/A | *pla* | 8374–8902[a] |
| cafpri_f | CCAGCCCGCAT CACT | cafpri_r | ATCTGTAAAGTTAA CAGATGTGCTAGT | Tqpro_caf | JOE-AGCGTACCAA CAAGTAATTCTGTA TCGATG-BHQ1 | *caf1* | 109–255[b] |
| Y.pes_pPCP_F | AGACATCCGG CTCACGTTAT | Y.pes_pPCP_R | GAGTACCTCCT TTGCCCTCA | Y.pes_pPCP_Pr | FAM-CACCTAA TGCCAAAGTCTTT GCGGA-TAMRA | *pla* | 8550–8669[a] |

**Notes.**
[a] The nucleotide position of the Y.pes_pPCP_F and Y.pes_pPCP_R primers based on the CP000310.1 *Yersinia pestis* Antiqua plasmid pPCP.
[b] The nucleotide position of the cafpri_f and cafpri_r primers based on the KF682424.1 *Yersinia pestis* strain S1 plasmid pMT1 capsule protein F1 (*caf1*) gene.

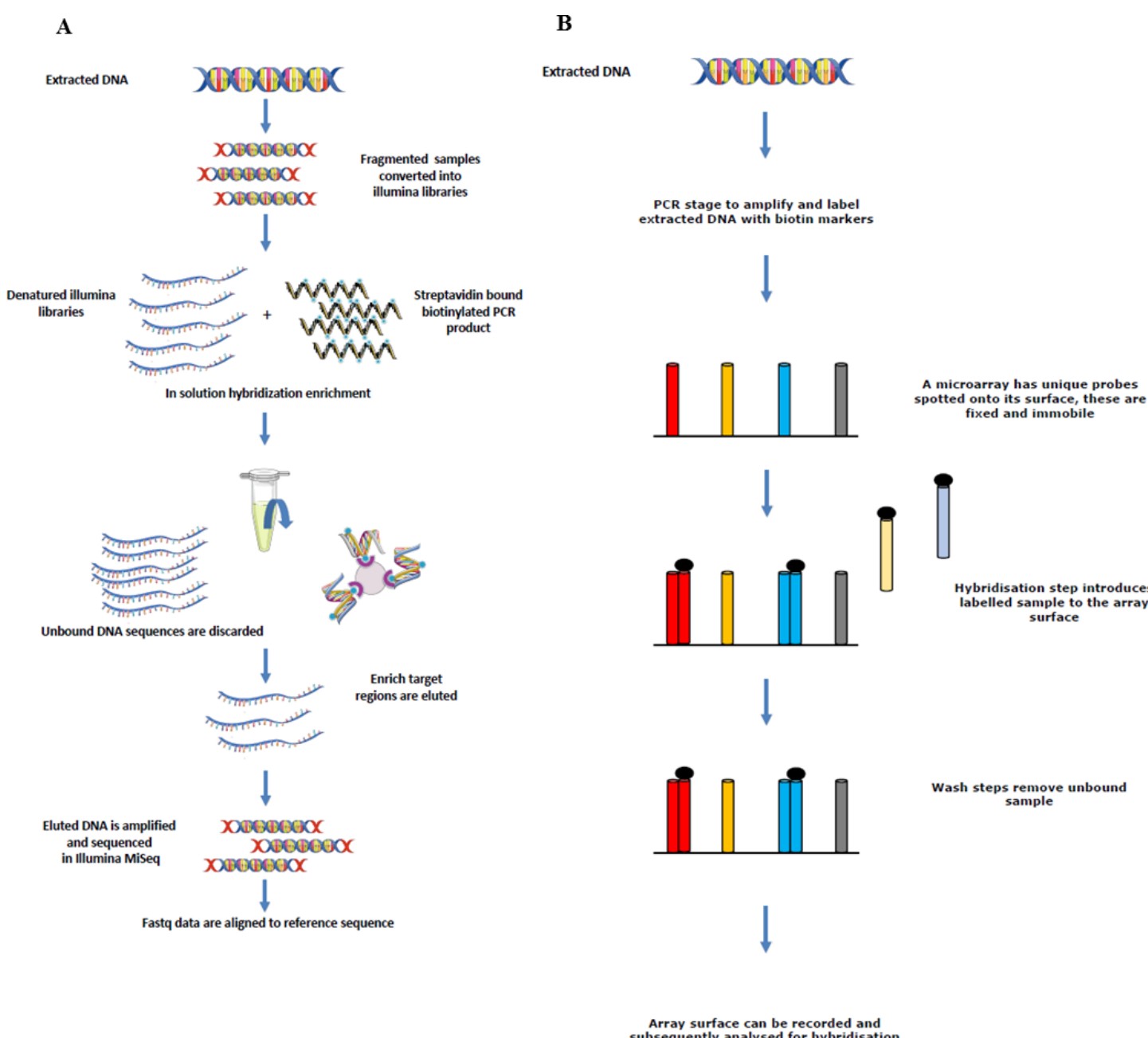

**Figure 1** Sequence showing how extracted DNA is used for sequence enrichment capture (A) and microarray hybridisation (B).

## KEY FINDINGS

1. Sequences homologous to the *pla* gene, which is present in *Y. pestis,* have been found in samples from several rodent species, in the absence of *Y. pestis.*
2. PCR, microarray and sequencing data suggest that these sequences may be present in environmental bacteria.

3. Caution is warranted in interpreting screening results for detection of *Y. pestis* when the *pla* gene is used as the sole marker for the presence of the pathogen.

## ACKNOWLEDGEMENTS

We would like to thank Chelsea Himsworth, Florence Ayral, and Kieran Pounder for providing the tissue samples. The authors would also like to thank Brendan Wren from the London School of Hygiene and Tropical Medicine for providing the *Y. pestis* DNA sample. Finally, the authors would like to thank Tom Giles from the University of Nottingham's Advanced Data Analysis Centre.

### Funding

This research has received funding from the European Union Seventh Framework Programme (FP7/2007-2013) under grant agreement no 222633. The funders had no role in study design, data collection and analysis, decision to publish, or preparation of the manuscript.

### Grant Disclosures

The following grant information was disclosed by the authors:
European Union Seventh Framework Programme: FP7/2007-2013.

### Competing Interests

The authors declare there are no competing interests.

### Author Contributions

- Timothy A. Giles conceived and designed the experiments, performed the experiments, analyzed the data, contributed reagents/materials/analysis tools, wrote the paper, prepared figures and/or tables, reviewed drafts of the paper.
- Alex D. Greenwood conceived and designed the experiments, reviewed drafts of the paper.
- Kyriakos Tsangaras conceived and designed the experiments, performed the experiments, analyzed the data, contributed reagents/materials/analysis tools, reviewed drafts of the paper.
- Tom C. Giles analyzed the data.
- Paul A. Barrow and Duncan Hannant reviewed drafts of the paper.
- Abu-Bakr Abu-Median conceived and designed the experiments, contributed reagents/materials/analysis tools, reviewed drafts of the paper.
- Lisa Yon conceived and designed the experiments, wrote the paper, reviewed drafts of the paper.

### Animal Ethics

The following information was supplied relating to ethical approvals (i.e., approving body and any reference numbers):

The School of Veterinary Medicine and Science approved this study by letter.

## Microarray Data Deposition

The following information was supplied regarding the deposition of microarray data:
GSE77765.

## Data Availability

The raw data is available from GEO GSE77765.

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
