# Peer review of "Detection of a Yersinia pestis gene homologue in rodent samples"

_PeerJ, doi:10.7717/peerj.2216_

## Round 0.1 · original submission · Minor Revisions

Dear Dr. Giles,

Please comply with all the suggestions of reviewers. Two of the comments: the possibility of bacterial Pla instead of rat one detected in your work; as well as introducing figure describing the various approaches taken to ID the Pla gene deserve special attention.

·

Basic reporting

The authors of the manuscript "Detection of a Yersinia pestis gene homologue in rodent samples" report the detection of a homologue of the pla gene in isolates from two rat and two mice species. pla is widely used to screen for the presence of Yersinia pestis, the causative agent of plague. In this study the authors used array-based and PCR-based screening approaches.
The observations reported here and in related work have to be considered when designing screening approaches for Yersinia pestis and when interpreting screening results.

I suggest to improve the readability of the manuscript by making a clearer distinction between 'Results' and 'Methods'. The 'Results' part could be condensed in order to give a concise summary of all findings without giving details on the applied methods. In addition, I suggest to put the 'Results' part right after the introduction. The 'Discussion' could be merged with the 'Results' part into a 'Results and Discussion' section.

I suggest to cite and consider Haensch et al. 2015 ('The pla gene, encoding plasminogen activator, is not specific to Yersinia pestis') in the context of this study.

As the pla gene encodes an important virulence factor I suggest to provide a bit more information on the known function of the encoded protein. In addition, some general information on the PCP1 plasmid should be provided.

The authors state that "future pandemics of Y. pestis are unlikely". I find it hard to estimate the probability of future Y. pestis pandemics. I suggest to weaken that statement a bit.

Experimental design

No Comments

Validity of the findings

No Comments

Reviewer 2 ·

Basic reporting

Overall the paper is really interesting, but I think it lacks a clearly defined goal or target. It appears a bit out of general context. I don't see major flaws in the reporting and is easily fixed with a few more bits and pieces of information.

Experimental design

Much of the specific research design and experimental outlay could use a good overhaul. A figure describing the various approaches taken to ID the PLA gene would make reading much easier and desirable. There is no sequence data or analysis of the sequence data post enrichment mentioned specifically. It may be worthwhile to show some of the metrics of the enrichment (success/failure) from the dataset. It was unclear of the sequence data produced additional read data from PLA or other Yp homologs that may have been captured.

Validity of the findings

Solid and of general interest.

I think, given my comment below and without further work, I would hesitate to stress that the homolog is clearly found with Rat. Could it not be of bacterial origin as well?

Additional comments

Well done! I would suggest the authors read, interpret and incorporate the following papers into the mix - I think it is important to note. Perhaps these PLA signatures are bacterial in origin after all, not rodent homologs?
The pla gene, encoding plasminogen activator, is not specific to Yersinia pestis
Stephanie Hänsch, Elisabetta Cilli, Giulio Catalano, Giorgio Gruppioni, Raffaella Bianucci, Nils C. Stenseth, Barbara Bramanti, and Mark J. Pallen, doi: 10.1186/s13104-015-1525-x

---

## Round 0.2 · accepted · Accept

The names of species should be in italics. Please standardize it in the References.

·

Basic reporting

The authors of the article "Detection of a Yersinia pestis gene homologue in
rodent samples" have submitted a revised version of their manuscript.

All suggestions of the reviewers have been addressed.
I have no further comments.

Experimental design

No Comments

Validity of the findings

No Comments